# TORCH Congenital Syndrome Infections in Central America’s Northern Triangle

**DOI:** 10.3390/microorganisms11020257

**Published:** 2023-01-19

**Authors:** Mary K. Lynn, M. Stanley Rodriguez Aquino, Stella C. W. Self, Mufaro Kanyangarara, Berry A. Campbell, Melissa S. Nolan

**Affiliations:** 1Department of Epidemiology and Biostatistics, Arnold School of Public Health, University of South Carolina, Columbia, SC 29208, USA; 2Centro de Investigación y Desarrollo en Salud, Universidad de El Salvador, San Salvador, El Salvador; 3School of Medicine, University of South Carolina, Columbia, SC 29208, USA

**Keywords:** TORCH congenital syndrome, TORCH Central America, congenital infections Central America, congenital syndromes Central America’s Northern Triangle

## Abstract

TORCH pathogens are a group of globally prevalent infectious agents that may cross the placental barrier, causing severe negative sequalae in neonates, including fetal death and lifelong morbidity. TORCH infections are classically defined by *Toxoplasma gondii*, other infectious causes of concern (e.g., syphilis, Zika virus, malaria, human immunodeficiency virus), rubella virus, cytomegalovirus, and herpes simplex viruses. Neonatal disorders and congenital birth defects are the leading causes of neonatal mortality in Central America’s Northern Triangle, yet little is known about TORCH congenital syndrome in this region. This review synthesizes the little that is known regarding the most salient TORCH infections among pregnant women and neonates in Central America’s Northern Triangle and highlights gaps in the literature that warrant further research. Due to the limited publicly available information, this review includes both peer-reviewed published literature and university professional degree theses. Further large-scale studies should be conducted to clarify the public health impact these infections in this world region.

## 1. Introduction

Maternal mortality in Central America’s Northern Triangle is high [1]. As of 2015, maternal mortality ratios in El Salvador, Guatemala, and Honduras per 100,000 live births were 54, 88, and 129, respectively [1]. Approximately 25% of women aged 20–24 years have their first child before the age of 18 [1]. Neonatal disorders and congenital birth defects comprise the leading causes of neonatal mortality [2]. Urban population growth over the past two decades in Central America has been twice that of other Latin American populations [3]. Compounding factors including urban population growth, income, and a unique ecological climate make infectious diseases prominent throughout the region [3]. Public health capacity to monitor vector-borne and other infectious diseases is challenging, particularly in underserved regions with limited resources, leading to under-detection and under-reporting, particularly in the most vulnerable populations.

Pregnant women in Central America’s Northern Triangle face a host of challenges, including restricted or limited education surrounding contraceptive use, access to prenatal care, high rates of gender-based violence, total abortion restrictions, and underage pregnancy [4,5]. Rural residing and indigenous persons comprise a large proportion of the Central American population, making access to care and culturally informed care even more challenging among these groups [4]. El Salvador is the only Central American country with a specific law to prioritize and promote maternal–child health throughout the perinatal period [6]. This law is aligned with World Health Organization (WHO) recommendations for postnatal care, promotion of breastfeeding and mother–child bonding [6].

TORCH infections are vertically transmitted and are infectious causes of considerable congenital and neonatal morbidity and mortality globally [7]. Traditionally, these infections include toxoplasmosis, other infections of concern, rubella, cytomegalovirus (CMV), and herpes simplex virus (HSV). The “other” category includes various other communicable pathogens, including hepatitis B and C, HIV, syphilis, Chagas disease, Zika virus, varicella, and parvovirus B19 [8,9]. Up to 30% of stillbirths are attributed to infectious causes, although further research is needed to clarify the burden of these infections globally [10]. The symptoms of TORCH congenital syndrome are vast, overlapping, and often severe, leading to lifetime morbidity (Table 1) [10].

Generally, TORCH infection clinical outcomes commonly include low birthweight, preterm birth, stillbirth, hearing and vision loss, and neurological and developmental sequalae that may have lifelong impacts on affected children [9,10]. Microcephaly, hydrocephalus, developmental anomalies and delays are other hallmarks of many TORCH pathogens [12]. The exact mechanisms of placental infection or otherwise vertical transmission pathways are not fully understood [7]. However, studies have focused on pathogen invasion of the trophoblast, decidua, maternal capillaries or other vasculature following immune damage to the placenta, or through placental macrophages [7,10]. The risk of infection and vertical transmission varies by pathogen and by world region [9]. Typically, neonatal outcomes are more severe when these infections are acquired in the early stages of pregnancy, early in fetal development [9]. Early detection and treatment (when available) are essential for preventing potential lifelong morbidity as well as maternal and neonatal mortality. TORCH infections disproportionately affect low-and middle-income countries, and limited resource capacity for surveillance leaves the true burden of these infections unknown [13]. This review will highlight traditional TORCH pathogens along with three additional salient pathogens responsible for perinatal infections in Central America’s Northern Triangle: Honduras, Guatemala, and El Salvador. In 2015, the Pan American Health Organization certified the WHO Region of the Americas as the first world region to eliminate rubella and rubella congenital syndrome, with zero cases reported in Central America’s Northern triangle after 2006 [14,15]. Therefore, rubella was excluded from this review.

The United Nations 2030 Agenda for Sustainable Development, Goals 3.1–3.3, aim to reduce maternal mortality, end preventable mortality among neonates and children <5 years, and combat communicable disease transmission [16]. Despite increasing international efforts to monitor the prevalence of congenital anomalies, Central America’s Northern Triangle is not included in any coordinated health agency efforts to monitor these outcomes [17]. Due to the importance of TORCH infections in the context of maternal child health and the associated severe lifetime morbidity among those infected, this literature review aimed to clarify what is known about TORCH infections and pertinent knowledge gaps in Central America’s Northern Triangle.

## 2. Materials and Methods

A literature review was conducted to synthesize the limited information regarding TORCH infections in Central America’s Northern Triangle as defined by the following infections: *Toxoplasmosis gondii*, *Trypanosoma cruzi*, Zika virus, Dengue virus, rubella virus, cytomegalovirus, and herpes simplex virus. We searched the following databases for any peer-reviewed articles in English or Spanish between the years 2012 and August 2022: Pubmed, Google Scholar, ScieLo, PAHO, Latin American and Caribbean Health Sciences Literature, MASCOT/Wotro Map of Maternal Health Research, and Repositorio Centroamericano SIIDCA-CSUCA. University repositories from El Salvador, Guatemala, and Honduras were searched for bachelor’s, master’s, and professional degree theses to further include studies not published in peer-reviewed scientific literature. Search terms were extensive, and the search was conducted with terms in both English and Spanish. Due to a general lack of information surrounding TORCH infections in this region, we included articles on the previously specified pathogens within the context of congenital infections, pregnant women, and women of childbearing age.

Reasons for exclusion from the final manuscript included: outside of date range, not original research, outside of target population, and outside of target countries of interest. Given the rarity of these infections within the common scientific literature, the overall majority of articles selected for this review were in the gray literature; consequently, the traditional PRISMA method was not employed. Findings from gray literature publications and other data sources used for this manuscript can be further found in Appendix A.

## 3. Results

### 3.1. Toxoplasmosis

*Toxoplasma gondii* is the etiologic agent of the highly prevalent zoonotic infection toxoplasmosis [18]. *T. gondii* is typically acquired environmentally through ingestion of sporulated oocysts in contaminated food or water [18,19]. Estimates suggest a prevalence of nearly 30% of the global human population, although prevalence varies widely between nations and world regions [20]. Although nearly half of infections are subclinical, severe infections can occur within the immunocompromised or in congenital cases, translating to 1.2 million disability-adjusted life years globally [21,22,23].

*T. gondii* is a genetically and geographically heterogenous parasite with three primary circulating strains (Type I, II, III), with Type I shown to be more virulent compared to other circulating strains [24,25]. Although few genotyping studies exist from Central America, Type I strains were identified in congenital toxoplasmosis cases in this region, with atypical strains found in sylvatic environments along with higher-than-average parasite strain diversity [20,26]. South American strains are varied with documented higher disease severity [27]. Differences in symptomology and symptom severity of congenital toxoplasmosis have been identified by world region, although the connection between strain type and clinical manifestations should be further explored [20].

Of primary clinical importance is vertical transmission that occurs when tachyzoites from an infected mother cross the placenta, infecting the fetus [25]. Consistent with other protozoan parasites, the direct mechanisms of congenital transmission and host–parasite interaction of *T. gondii* with the placental barrier are not well understood [28]. Immune response and tissue damage caused by pro-inflammatory cytokines are thought to be contributors [29]. Vertical transmission primarily occurs among pregnant women with acute, primary infection during pregnancy, and less commonly, with infection just prior to gestation [20,30]. Among chronically infected women, congenital transmission may occur from reactivation of the parasite if immunocompromised or reinfection with a more virulent strain [20,30]. Transmission risk for the *T. gondii* parasite varies across maternal pregnancy stages, with increasing risk across gestational age [21]. Changes in maternal immune factors along with decreasing anatomical thickness of the placenta are thought to play a major role in this increasing risk over the course of pregnancy [20,21,30]. Contrastingly, congenital infection severity and risk of symptomatic congenital infection are higher when infection occurs earlier in pregnancy [21]. Parasite load, *T. gondii* strain virulence, and maternal immune function also contribute to congenital disease severity [23]. Due to variations in the seroprevalence of *T. gondii* infection among women of childbearing age, the risk of vertical transmission varies by country and world region [31]. Transmission risk further depends on maternal factors, including maternal comorbidities and treatment within the prenatal period [20]. The overall risk of vertical transmission among primary-infected mothers ranges from 20–50% [30]. However, third trimester infections have shown rates of vertical transmission between 60 and 80% [32].

Clinical manifestations of congenital toxoplasmosis vary, including microcephaly, hepatosplenomegaly, jaundice, petechiae, thrombocytopenia, anemia, transaminitis, seizures, hearing and vision loss, and developmental delay [29,30,33]. Typical hallmarks of infection include intracranial calcification, hydrocephalus, and choroidoretinitis [27], although hydrocephalus occurs in only 4% of symptomatic congenital infections [21]. In the absence of congenital infection, negative birth outcomes may also result when mothers are chronically infected, as slower fetal development and postnatal developmental delay have been associated with chronic maternal infection [22].

Treatment with spiramycin or a pyrimethamine and sulphonamide combination is recommended for primary acute infection among pregnant women to prevent maternal-fetal transmission; however, it is still unclear whether treatment reduces transmission risk, as no randomized control trials have thus far been conducted [20,27]. While most published studies show benefits in some clinical respects, randomized control trials are needed to assess treatment efficacy at various intervention time points during acutely infected pregnancies.

Ultrasound may detect hallmark symptoms of congenital *T. gondii* during pregnancy along with intrauterine growth restriction and hepatosplenomegaly, prompting clinical suspicion for testing [27]. However, severe congenital infections are primarily diagnosed through routine neonatal screening, when neurological or ophthalmological symptoms present along with maculopapular rash, jaundice, and small for gestational age [23,34]. Following clinical suspicion and confirmation of maternal infection, *T. gondii* serology is performed [23]. IgG avidity testing is further conducted to assess temporality of infection and risk to the fetus, particularly up to 16 weeks, to exclude acute infection if there are high avidity results [20]. Molecular methods using amniotic fluid, peripheral blood, urine, or cerebrospinal fluid PCR also confirm congenital transmission [23].

Screening of pregnant women during prenatal care has been implemented in few countries, primarily in Europe, where programs are both clinically beneficial and cost-effective [34,35]. Screening programs are a critical component of public health intervention, as pregnant women or women planning to become pregnant may have the opportunity to avoid unnecessary behaviors that increase the risk of primary infection (poor sanitation, unhygienic practices, eating undercooked meat) [22]. Further, acutely positive women may have the opportunity for prophylaxis and treatment intervention to prevent parasite transmission to the fetus.

Congenital *T. gondii* studies and seroprevalence studies among women of childbearing age are limited in Central America’s Northern Triangle; however, the burden of latent infection is hypothesized to be high due to a well-established link between infection and low socioeconomic status (SES) or countries with a low human development index score [22]. Previous studies have attempted to quantify the global burden of both latent and acute *T. gondii* in pregnant women [22,31,33]. In El Salvador and Guatemala, the prevalence of *T. gondii* IgM among pregnant women is about 0.9% (range: 0.8–1.0%), while in Honduras, active infection is estimated at 1.0% (range: 1.0–1.1%) [31]. Seroprevalence of *T. gondii* IgG in El Salvador, Guatemala, and Honduras are estimated at 52.1% (range: 51.7–52.4%), 52.8% (range: 52.5–53.2%), and 29.7% (range: 29.4–30.0%), respectively [31].

In Guatemala, a study from San Juan Sacatepéquez evaluated two groups of 30 women of childbearing age with and without domestic pet cats for the presence of *T. gondii* IgM and found that no (N = 0) women were positive by chemiluminescent assay [36]. Another study in the Zacapa department, Guatemala, found 50.4% of women of childbearing age seropositive for *T. gondii* IgG [37]. Seropositivity was significantly associated with having an earthen floor in the home and varied across municipality and age group. Approximately 62% (N = 5/8) of participating pregnant women had positive IgG serology results [37]. A separate multi-site study found only 1 congenital toxoplasmosis case out of 523 of neonates from reference hospitals across three departments [38]. The resulting seropositive neonate was born premature at 35 weeks of gestations at 5 pounds with no other remarkable symptoms noted [38]. In Honduras, one study found 48% *T. gondii* IgM positivity and 49% IgG positivity among 88 pregnant women attending prenatal care in Paraiso. The study further found a significant association with seropositivity and the risk of miscarriage [39]. A significant association was also observed between IgM positivity and lack of hand washing, keeping pets indoors, and ensuring food is well cooked before eating [39]. In El Salvador, one study found 26.5% *T. gondii* IgG (N = 26/98) seropositive among women of childbearing age [40]. None of the participants were positive for IgM, and 65% were pregnant (N = 34/98); however, the number of IgG seropositive pregnant women was not reported [40]. One thesis found IgG seroprevalence of 17.4% among 86 women from the departmental referent hospital blood bank in San Miguel, and zero were IgM positive [41]. A two-year study of microcephalic infants <1 year presenting to the national referent children’s hospital in 2015 and 2016 revealed three infants positive for toxoplasmosis and three infants with toxoplasmosis and cytomegalovirus coinfection [42]. Prior hospital studies found 15.3% of 85 pregnant women in Hospital Nacional de Usulután positive for acute infection [41]. In this same department in 2019, a suspected case of congenital toxoplasmosis resulted in infant death [43].

### 3.2. Other—Chagas Diseases

*Trypanosoma cruzi* is a kinetoplastid parasite and the infectious agent responsible for Chagas disease [44]. Due to globalization and population movement, imported human cases are now found worldwide with approximately 6 million globally infected and 71 million at risk for infection [45,46]. Vector-borne stercorarian transmission is limited from North America to the southern cone countries of Latin America due to the ecological habitat of the triatomine disease vector [45,47].

*T. cruzi* parasite has a highly complex plastic genome and is organized into seven sub-clades or DTU from TcI-TcVI, with six infecting humans and mammals (DTU TcI-TcVI) and a seventh distinct TcBat that circulates within *Chiropteran* populations [45,47,48,49]. Differential distributions of parasite DTU occur geographically, with the most common DTU, TcI, predominating from North America to the northern region of South America [50]. TCV and TcVI dominate within human Chagas disease transmission cycles in South American southern cone countries [47]. The influence of parasite DTU on vertical transmission is not well understood; however, some studies show DTU-dependent placental tropism [44,46].

Congenital transmission occurs through trypomastigote passage across the placenta for acute infection in the neonate [51]. Similar to other protozoans, *T. cruzi* is able to evade host immune response and cross the placental barrier; however, the distinct mechanisms of congenital transmission have yet to be clarified [46]. Susceptibility for vertical transmission depends upon virulence factors of the *T. cruzi* strain, level of parasitemia, ability of the parasite to evade immune response, and maternal and fetal immune responses to the pathogen [44,50]. High parasitemia is thought to contribute to destruction of the placental trophoblast layer, for differentiation within the basal laminae and stromal cells [46]. Therefore, vertical transmission is more likely to occur during acute or reactivated infection during pregnancy or when maternal parasitemia is significantly elevated in chronically infected mothers [50]. When parasitemia is lower, placental breaches or tears are thought to contribute to parasite passage of the placental barrier [50].

Nearly 60% of congenitally infected neonates are asymptomatic at birth; however, infected neonates may still progress to dilated cardiomyopathy and other typical chronic manifestations throughout their life [46]. Those that do present with symptoms may experience low birth weight, prematurity, anemia, myocarditis, hepatosplenomegaly, meningoencephalitis, respiratory distress syndrome, organomegaly, and low Apgar scores [45,46,50]. Neonatal death and spontaneous abortion may also occur in a small proportion of vertically transmitted pregnancies [45]. Adverse health outcomes are more likely to occur in cases of acute or reactivated maternal infection [50].

Gold standard testing for congenital infection is conducted through molecular methods such as PCR using primers from satellite or minicircle-derived kinetoplast DNA [45]. Resource limitations in certain endemic countries provide challenges employing molecular methods for disease diagnosis [46]. Therefore diagnosis of congenital infection is typically made by direct parasitological examination from birth to between 60 and 90 days coupled with serology 9 to 12 months of age in infants, after maternal antibodies have dissipated within the infant [45,46].

Although originally developed in the 1960s and 1970s, benznidazole and nifurtimox remain first line treatments for Chagas disease globally [45]. Anti-parasitic treatment among infants <1 year of age is highly effective, and common side effects are well tolerated among children [45].

Nearly 25% of the global burden of Chagas disease is due to congenital transmission [45]. The WHO estimates that within endemic countries, over 1 million women of childbearing age are infected with *T. cruzi*, resulting in approximately 9000 congenitally infected live births [52,53]. Overall, Chagas disease prevalence ranges between 5% and 40% among women of childbearing age across Latin America [51]. However, few large-scale population studies have been conducted in Central America, leaving the true burden unknown [51,54,55]. Vertical *T. cruzi* transmission occurs in approximately 5% of births to infected women but is highly variable by endemic country [51]. Further, the 5% overall congenital transmission rate is largely based on South American studies, where different parasite strains circulate [56].

Increased risk for vertical transmission has been linked to maternal poverty, malnutrition, and the presence of maternal comorbidities due to linkages with maternal immunomodulation [50]. In Guatemala, national *T. cruzi* prevalence is approximately 1.23% [55,57]. However, studies conducted between 2014 and 2015 in Jutiapa and Jalapa departments found between 3.3 and 10.8% *T. cruzi* prevalence among women of childbearing age [58,59]. Statistically significant associations between seropositive and age were observed [59]. A 2016 study in Jutiapa department found 3.8% seroprevalence among pregnant women [57]. A recent professional degree study in Chiquimula department found 27.6% seropositivity among women of childbearing age (N = 37/134), highlighting the variability and clustering of these infections in high-risk areas [60]. In Honduras, the prevalence of Chagas among all age groups is 0.92% [57]. Among women of childbearing age, the Honduran Ministry of Health reported 1% seropositivity within the years 2015 and 2017 [57]. One professional degree thesis found 1.6% (N = 46/2794) seropositivity among pregnant women across 12 departments between 2013 and 2015 [61]. The first published study of vertical transmission in Honduras was conducted in 2018. Of the 6583 women recruited at labor and delivery, 230 cord blood samples were positive by at least one rapid test, and 181 women were seropositive [56] At birth, 2.2% (N = 4/181) neonates were positive by direct parasitological examination at the time of birth, although zero cases of congenital transmission were identified when infants were screened at 10 months of age [56]. The first case report of confirmed congenital transmission was published shortly following, in 2019 [62]. Infection prevalence of *T. cruzi* in El Salvador is estimated to be 1.3% among all ages [55,57]. The first and only published study, to the best of our knowledge, regarding congenital Chagas transmission in El Salvador in 2015 revealed 3.8% (N = 36/943) seroprevalence among pregnant women from three high-risk departments [54]. One of the thirty-two neonates resulting from this study was positive and was documented as the first congenital Chagas case in El Salvador [54]. Previous professional degree studies found 8.6% (N = 10/116) seropositivity among pregnant participants from three health centers in eastern departments and 1.9% (N = 1/52) of third-trimester pregnant women seropositive with no evidence of vertical transmission [63,64]. Screening of pregnant women for *T. cruzi* incrementally began as part of recent technical guidelines in El Salvador, with a 0.44% seropositivity rate among 18,000 screened in 2018 [57]. However, the implementation of this program after 2018 is unknown and is absent from Salvadoran clinical guidelines for prenatal screening, and no data are publicly available [6,65].

### 3.3. Other—Zika Virus

Zika virus (ZIKV) is a positive-sense, single-stranded RNA virus of the family *Flaviviridae*, genus *Flavivirus*, that causes severe congenital defects when acquired during pregnancy [66,67]. ZIKV is principally transmitted via infected female *Aedes* mosquitoes and can also be sexually transmitted [3,12,68]. Due to the disproportionate frequency of microcephaly cases produced during the 2015–2016 outbreak, ZIKV was the first arbovirus to be linked to teratogenic birth anomalies [68,69,70]. Although vertical and sexual transmission had been reported in prior outbreaks in French Polynesia, the more recent South American epidemic provoked the WHO to confirm causal associations between ZIKV virus and congenital abnormalities for the first time in March 2016 [7,68].

The primary pathway of concern is vertical transmission through the placenta, with severe manifestations of congenital Zika syndrome occurring when maternal infection is acquired early in pregnancy [12]. ZIKV has also been detected in breast milk, saliva, and urine, creating the potential for additional neonatal transmission outside of the traditional placental pathway [12]. Risk of vertical transmission is dependent upon a variety of factors including timing of maternal infection, comorbidities, host immune response, and serostatus for other flavivirus infections [10]. Although exact mechanisms remain unclear, non-structural proteins NS4A and NS4B are thought to be involved in enhanced autophagy, causing disrupted centrosome stability, contributing to hallmark central nervous system abnormalities [7,71]. Maternal inflammatory response induced trophoblast and fetal capillary damage along with viral targeting of progenitor cells, and cortical thinning also contributes to vertical transmission risk [7]. The impact of antigenic cross-reactivity with other flaviviruses is particularly important in congenital transmission but has not been fully clarified [70]. Studies have shown varying results with both protective, enhanced host immune response and antibody-dependent enhancement activation leading to increased placental pathogenesis and viral invasion of the trophoblast [70].

In newborns, congenital Zika syndrome results in the hallmark of microcephaly or other varying degrees of brain malformations and neurologic sequelae [67]. Rates of vertical transmission and resulting congenital manifestations in infected neonates have been difficult to assess, as rigorous study designs are challenging to employ in the context of congenital Zika transmission [7,12,72]. Likely depending on first or early second trimester gestational age when maternal infection occurs, studies have estimated risk anywhere between 1 and 13% [7,12,73]. First trimester infections have also been associated with abortion, stillbirth and neonatal death [7]. Other congenital infection symptoms include brain and ocular anomalies, cranial abnormalities, and neurological damage that lasts, in most cases, throughout the lifetime of the child [67,74]. Congenital Zika virus may also result in mortality with a median rate of 41 per 1000 infected neonates; however, data are based on limited studies with small sample sizes [67].

Congenital Zika may be detected early in pregnancy if central nervous system abnormalities are observed via ultrasound and are confirmed via positive serologic maternal test along with quantitative RT-PCR in amniotic fluid or neonatal serum [12]. Case reports have retrospectively detected ZIKV in placental and brain tissue as well as in cerebrospinal fluid following stillbirth [12].

In both the general population and among vulnerable groups such as pregnant women, diagnosis, treatment, and prevention are challenging. Antiviral treatments and vaccine candidates are not yet available for ZIKV infection [7]. Therefore, clinical management is typically palliative in nature.

Due to overlapping unique climatic, social, and ecological factors including land use change, deforestation, and urban population growth, Central America is particularly vulnerable to mosquito-borne disease emergence and resurgence [3]. The first locally acquired ZIKV cases were identified in November 2015 in El Salvador and Guatemala and have since established endemicity, although cases burdens have lessened since the initial outbreak [3]. The majority of studies regarding vertically transmitted ZIKV have been conducted in mouse models, with more rigorous studies concentrated in Brazil and Colombia [12]. From May 2015 to January 2018, the Pan American Health Organization recorded cases of ZIKV and congenital Zika syndrome, confirmed by the WHO case definition. Between these dates, El Salvador reported 11,789 cases of ZIKV with 51 confirmed, along with 4 confirmed congenital cases. Guatemala reported 3907 cases with 1032 confirmed cases and 140 confirmed congenital cases. Honduras reported 32,385 cumulative cases with 308 confirmed and 8 instances of zika congenital syndrome [75]. According to PAHO, current ZIKV cumulative incidence in El Salvador, Guatemala, and Honduras as of 2 August 2022 is 1.96, 8.01, and 0.16 per 100,000 population, respectively [76].

Due to limitations in access to care for underserved populations and the proportion of asymptomatic infections among the general population alone, the prevalence of ZIKV among pregnant women in unknown. No studies from the published or gray literature were obtained from Honduras or Guatemala, as most of the Central American research regarding congenital Zika syndrome has been centered in Nicaragua, Panama, and Costa Rica. In El Salvador, when symptomatic women presenting with symptoms are diagnosed, cases are routinely reported to the Ministry of Health, and data are publicly available. In 2016, the Ministry of Health published technical guidance for infants with microcephaly in public hospitals, resulting in several small-scale studies at the University of El Salvador. One such study at the national reference hospital, Hospital Nacional de Niños Benjamín Bloom (HNNBB), found 26.8% of a cohort of microcephalic infants had mothers with diagnosed ZIKA during pregnancy, and 7.3% of infants were positive for ZIKV [42]. Underserved populations with lower access to health care have been associated with higher risk of ZIKV infection [12]. The proportion of asymptomatic infections coupled with the high cross-reactivity to other flaviviruses along with the inability to eradicate the highly urbanized *Aedes* vector species warrants continued ZIKV research and rigorous screening guidelines among pregnant women to prevent congenital anomalies [77].

### 3.4. Other—Dengue Virus

Dengue virus (DENV) is a widely circulating positive-sense, single-stranded RNA flavivirus that has increased in incidence 30-fold over the last 50 years [78,79,80,81]. Nearly 53% of the world population is at risk for infection [82]. Considered the most prevalent arboviral infection, the large rise in incidence over recent decades has further been accompanied by more frequent outbreaks of increased intensity and geographical range [80,81,83]. Nearly 400 million cases of infection, 500,000 hospitalizations, and 20,000 annual fatalities are attributable to DENV globally each year [81]. Transmitted by *Aedes* mosquitoes, dengue virus is divided into four distinct serotypes (DENV-1, DENV-2, DENV-3, DENV-4) [84]. Specific neutralizing antibodies confer lifetime immunity to the infecting serotype [82]. Antibody-dependent enhancement in secondary infections contributes to a higher risk of severe dengue; however, pathogenic mechanisms of dengue are complex, and gaps exist in the understanding of exact viral mechanisms leading to risk for severe infections [84]. In many endemic countries, all four serotypes circulate concurrently. Differences in virulence have also been found between DENV serotypes and world regions, with Southeast Asian DENV-2 strains implicated in the increased pathogenicity seen in past Latin American outbreaks [84].

DENV may also be transmitted vertically through the placenta or through breastmilk, although accurate estimates of vertical transmission rates and the prevalence of dengue among pregnant women have been limited by common asymptomatic presentation and limitations in study designs of the current literature [78,85,86]. Symptoms typically associated with dengue including thrombocytopenia and vascular permeability may contribute to viral passage across the placental barrier, and initiation of pro-inflammatory-induced placental damage may further lead to vertical transmission along with placental and fetal damage [87]. However, vertical transmission mechanisms remain unclear given the limited information in general surrounding perinatal and congenital dengue.

Typically, the risk of severe dengue increases with the second infection with a new serotype, but it may occur with primary infection, especially in neonates born to seropositive mothers [84]. Increased risk for severe dengue among maternal infections has also been associated with prior Zika virus infection, age, comorbidities, viral load, and viral strain of secondary infection [82].

Major gaps exist in the understanding of clinical implications during pregnancy and birth outcomes of the respective neonate [78]. Existing studies have found evidence suggesting that severe infections and the risk of medical intervention are higher among pregnant women [78]. Severe hemorrhage, shock, emergency C-section, and thrombocytopenia requiring transfusion have been reported along with approximately three times the risk of stillbirth and maternal mortality [68,85,88,89]. Risk of miscarriage, intrauterine growth restriction, placental abruption, and low-birthweight infants have been frequently reported [86,87,88].

The few studies from the literature surrounding congenital infection found, among symptomatic pregnancies, increases in risk of neurological congenital defects up to 50%, when maternal infection presented with symptoms, along with four times the risk for central nervous system defects, which have been reported in population-based studies from Brazil [19]. The rate of symptomatic infections among vertically infected neonates is also not well understood and mainly relies on case reports: severe dengue, dengue hemorrhagic fever, and dengue shock syndrome have all been reported [68,88]. Timing of infection, particularly directly before delivery, and severity of maternal infection are thought to be critical in maternal and neonatal outcomes [68,89].

Within symptomatic febrile onset, nucleic acid amplification tests (NAAT) and RT-PCR are commonly used to identify infections. Serological diagnostics for targeting the NS1 noncoding protein antigens can further be employed after approximately day 5 of symptom onset [81,82]. However, these assays may be cross-reactive with other flaviviruses, such as ZIKV. Subsequent infections in those previously positive for flaviviruses may mount low IgM and elevated IgG response against various flaviviruses, making serologic methods challenging [82]. PRNT tests provide more specific quantification of IgG that may help combat false positives and distinguish between flaviviruses [82]. Diagnosis of congenital infections is performed using qRT-PCR when available or serologic testing, following neonatal symptom presentation and confirmation of infection in the mother [85,90,91]. Placental tissue and cord blood have also been used to detect cases of vertical transmission [85].

Although the Dengvaxia DENV vaccine is available in many countries, safety has not yet been evaluated for pregnant women, and no antivirals exist for specific DENV treatment [81,82]. However, clinical intervention is necessary in severe cases to address blood and fluid loss as well as multiorgan involvement. Supportive care can reduce morbidity, as studies have reported nearly a 13% case fatality rate among severe dengue patients [82].

The prevalence of dengue among pregnant women and the risk of vertical transmission during pregnancy is not well understood [78]. Infection is often asymptomatic, and complications arising during pregnancy are not easily distinguished from other perinatal conditions such as eclampsia and hemolysis-elevated liver enzymes and low platelet count syndrome (HELLP) [68,78,91]. The rate of vertical transmission varies by study between 15% and 23%, to up to 90% in a case series of severely infected mothers during a large outbreak [68,91]. No case reports or other studies regarding dengue virus in pregnancy were found during this literature search for countries in Central America’s Northern Triangle, including studies from the gray literature, highlighting the need for further studies surrounding dengue in the peripartum period and vertical transmission in this highly endemic world region.

### 3.5. Cytomegalovirus

Cytomegalovirus (CMV), or betaherpesvirus and human herpesvirus 5, is a globally prevalent double-stranded DNA virus [92,93]. It is one of the nine herpesviruses that circulate among human populations and is the most common congenital infection worldwide [93,94,95]. Virions are composed of an approximately 235 kb linear genome, coding for >160 gene products, making it the largest among herpesviruses [92,93,96]. A large proportion of encoded genes play a critical role in host immune evasion, viral latency, and extensive cellular tropism, enabling the pathogen to infect a broad range of cell types [96]. Viral encoding of many human inflammatory protein homologs suggests co-evolution of the virus, which may contribute to its reach among humans and ability to widely persist in the global population [96].

CMV is transmitted person to person when saliva from an infected person passes through mucosal membranes, and it can also be transmitted sexually by blood transfusion, transplant, and vertically through the placenta [97]. Seropositive persons may further transmit the infection through other body fluids such as semen, breastmilk, urine, and vaginal secretions. Both primary infection during pregnancy and non-primary infection, reactivation of latent infection or infection with a new strain may result in congenital infection [98]. Primary infections, particularly within the first and early second trimesters, result more frequently in symptomatic congenital infections [92]. While reactivated CMV transmission is uncommon, non-primary infection results in a higher burden of congenital infection overall due to the high prevalence of CMV globally [10,97,98,99]. Transmission risk to the developing fetus increases throughout the stages of pregnancy and is approximately 40% overall for primary infection [10]. Mechanisms of vertical transmission have not fully been clarified, and they have been mostly investigated through human placental explants and cell cultures [10].

In the absence of underlying comorbidities, 80% of primarily infected pregnant women will show no symptoms [92]. Nearly 90% of infected neonates are subclinical at birth; however, 15% of asymptomatic, infected infants will later develop sensorineural hearing loss, retinitis, and cognitive impairment [92,98]. The 10% of neonates symptomatic at the time of birth develop disease manifestations including jaundice, “blueberry muffin” rash, hepatosplenomegaly, petechiae, microcephaly, intrauterine growth restriction, retinitis, and optic atrophy [92,93]. Sensorineural hearing loss is the hallmark of congenital CMV and is the most common nongenetic cause of hearing loss [97]. Developmental disorders, cognitive delay, and hearing and vision loss persist in up to 40% of infants symptomatic at birth [92,93].

Detecting infection within the perinatal period is challenging due to the absence of symptomatic infection and lack of maternal screening guidelines in most world regions. Congenital infections are diagnosed based on the timing and presence of clinical suspicion. Following abnormal ultrasound findings, PCR can be performed on amniotic fluid. In neonates, classic symptoms and neonatal hearing abnormalities are present. PCR is typically performed up to 3 weeks of life using saliva or urine [97,98].

Treatment of moderate to severe symptomatic, congenitally infected infants with valganciclovir is recommended [93,100]. Since the 1970s, many attempts to create a CMV vaccine have been undertaken with multiple, current Phase I and II clinical trials; however, none have thus far been approved [93]. Serial screening before 14 weeks of pregnancy is useful to identify primary infection in previously seronegative women. This method has limited utility among those already seropositive [97]. Infection prevention mainly relies on maternal education and improved hygiene practices among women with small children, as 25% of infected infants shed the virus in saliva and urine for approximately 18 months [92,101]. Infection among pregnant women often occurs via contact with an infected child <3 years of age in the family or via childcare [101]. Several studies show that improved hygiene practices significantly reduced maternal infection in the presence of infected infants at home in the United States and Europe [101].

Seroprevalence of CMV infection is globally variable, with estimates in some world regions among women of reproductive age reaching as high as 95% [95]. In the WHO Americas region, it is estimated that 79% of women of childbearing age are infected, and 87% of Salvadoran women of childbearing age have latent infection [95]. Estimates of seroprevalence of CMV among women of childbearing age and among the general population were unavailable for Guatemala and Honduras. No studies from the gray literature were obtained for these countries, as most research surrounding congenital CMV were available only from Costa Rica and Nicaragua. Although there is no national surveillance in El Salvador, limited studies have revealed that congenital transmission of CMV is a public health issue in this country. One study at the national referent hospital found 5 out of 41 (12.2%) infants with microcephaly presenting to the national referent hospital were positive for CMV infection, and 3 were coinfected with both CMV and toxoplasmosis [42].

### 3.6. Herpes Simplex Virus

Herpes simplex virus (HSV) is a double-stranded human herpesvirus affecting the facial, oral, and genital areas [102]. Responsible for infection in up to 3% of pregnant women globally, HSV is transmitted through direct contact with body fluids, lesions, or secretions and induces lifelong infection in the host [103]. HSV-1 and HSV-2, the two major subtypes globally circulating in humans, are approximately 40% homologous [102].

General transmission typically occurs through viral shedding from both asymptomatic and symptomatically infected persons [102]. Viral DNA can be shed between 6 and 12 days of infection, reaching the trigeminal or lumbosacral ganglia, establishing latency and persisting throughout life [102]. Immunomodulation and stress then trigger reactivation, although genital HSV-2 reactivation episodes tend to lessen in frequency after 3 months post-primary infection [102]. Viral shedding is shorter among asymptomatic persons, and between 0.4% and 1.4% of pregnant women shed the virus genitally at the time of delivery [102]. Up to 90% of all primary and recurrent maternal infections present asymptomatically [102]. Generalized malaise and headache are most commonly displayed, and about 50% will present with the hallmark prodrome along with localized pain and itching at the site of the skin lesion [102].

First trimester infections can result in vertical transmission of HSV across the placenta; however, this transmission mechanism is rare [104]. Transmission to the fetus can also occur in utero in approximately 5% of cases [104]. The majority of neonatal infections are caused when primary maternal infection occurs in the third trimester, close to labor/delivery and before maternal IgG mounts sufficiently for fetal protection [105]. These 85% of neonatal infections occur when HSV is shed in the genital tract and neonates are infected through contact with the birth canal [102,104,106].

Congenital and neonatal infections can be severe if untreated, with an estimated case fatality rate of 60% [106]. Congenital symptoms of HSV are similar to those of congenital Zika syndrome with microcephaly, retinitis, and limb abnormalities [104]. Clinical manifestations of neonatal infections may not develop until 3 weeks of life, which include central nervous system abnormalities along with skin, eyes, and mouth disease (SEM) that occur in nearly half of all symptomatic cases [106]. Encephalitis occurs in about one-third of symptomatic infections, and multiorgan disseminated disease occurs in about one-quarter of neonatal infections [104].

Clinical suspicion of HSV depends primarily on the presence of skin lesions on the face or genital tract, or through hallmark symptoms of neonatal disease. Due to cross-reactivity between the viruses, viral culture is the gold standard for HSV diagnosis; however, immunofluorescence and PCR are also used [102]. Serologic methods may be employed but do not differentiate between subtypes. Acyclovir can be used in pregnancy to prevent congenital and neonatal infection and may also be given in neonatal infections for up to 6 months of life to improve neurologic sequelae [104].

The global neonatal herpes risk is rare, occurring in every 1 out of 3000 to 20,000 live births globally. Most cases are attributed to HSV-2 infection [104]. Up to 30% of pregnant women are seropositive for HSV, and nearly one-tenth of women seronegative for both subtypes have seropositive sexual partners, leading to a seroconversion risk of 3.7% during pregnancy [102,105]. For those seronegative to HSV-2, there is a 2% chance of seroconversion during pregnancy [105]. However, transmission risk varies greatly between countries based on the number of sexual partners, socioeconomic status, education, contraceptive use, and other sexually transmitted coinfections [102]. However, one recent study estimated that the WHO Region of the Americas has the highest overall rate of neonatal herpes with 19.9 of 100,000 live births, and HSV-1 is thought to cause more cases of neonatal infection in this region [106]. Due to the presumed rarity of neonatal herpes infection, zero studies from the scientific or gray literature addressed congenital and neonatal HSV infection in Central America’s Northern Triangle.

## 4. Conclusions

The WHO region of the Americas became the first to eliminate rubella virus and congenital rubella syndrome as a public health problem in 2015. The remaining TORCH infections of public health concern in Central America have no available vaccines and limited preventive treatment to reduce the risk of maternal passage to the fetus or neonate. Maternal and neonatal screening programs for TORCH infections vary by country and are particularly difficult to standardize and employ across low-resource settings.

In addition to the challenges in capacity to monitor and screen women for these infections, Central American women are a uniquely vulnerable population, particularly those of low socioeconomic status, indigenous women, and those living in rural areas [107]. Central Americans are the second largest Latin American immigrant group in the United States [108]. Drivers of the high degree of migration out of this world region including sociopolitical conditions, poverty, violence, women’s personal agency and opportunity further contribute to the health status of women in this region [109]. Honduras has the highest rate of femicide among countries in Latin America and the Caribbean with 4.6 cases per 100,000 women, and El Salvador has the third highest rate with 2.7 cases per 100,000 women [110]. The unique circumstances among women in Central America’s Northern Triangle, combined with climatic and ecological conditions, and differentially circulating pathogen strains, warrant further research of TORCH infections in this region. The novelty of this manuscript is that we present information from the gray literature—an often overlooked, but valuable resource, in regions that culturally do not publish in the scientific literature often due to lack of financial or administrative resources. While not officially peer-reviewed, such studies are a leading source of information in these countries. All of the cited gray literature originates from scholarly theses published under the local university library system and may add value to the limited knowledge of TORCH infections in this region.

In order to reduce the often-lifelong morbidity of TORCH-affected children, as well as the mortality of pregnant women and neonates, future studies are necessary to inform clinical guidelines and to improve maternal child health in Central America’s Northern Triangle. United States-based obstetric–gynecologic and family medicine physicians within the maternal–child health space may benefit from a clarified understanding of these infections among the largest immigrant populations to inform patient care from diverse backgrounds.

## Figures and Tables

**Table 1 microorganisms-11-00257-t001:** TORCH pathogens and corresponding typical congenital/neonatal infection manifestations *^,1^.

Symptoms	Symptom Category	TOXO	CHAG	ZIKV	DENV	RUBV	CMV	HSV
Fever	Acute/Inflammatory condition				X			
Jaundice	Acute/Inflammatory condition	X				X	X	
Meningitis	Acute/Inflammatory condition		X		X			X
Rash	Acute/Inflammatory condition	X				X	X	X
Anemia	Blood disorder	X	X				X	
Thrombocytopenia	Blood disorder	X			X		X	
Hydrocephalus	CNS disorder	X				X	X	X
Intracranial calcifications/brain malformations	CNS disorder	X		X	X		X	X
Microcephaly	CNS disorder	X		X			X	X
Ventriculomegaly	CNS disorder	X		X			X	X
Low birthweight/IUGR	Growth/gestation impacts	X	X	X	X	X	X	
Preterm birth	Growth/gestation impacts		X		X			
Stillbirth/fetal loss/neonatal death	Growth/gestation impacts		X	X	X		X	
Chorioretinitis/ocular disease	Long-term sequelae/CNS disorder	X		X		X	X	X
Developmental delay	Long-term sequelae/CNS disorder	X		X			X	
Hearing loss	Long-term sequelae/Organ system conditions	X				X	X	X
Hepatosplenomegaly	Conditions of organs/tissues	X	X	X		X	X	X
Hydrops	Conditions of organs/tissues		X				X	
Cardiomyopathy/cardiaclesions/myocarditis	Conditions of organs/tissues		X			X	X	


* Toxo—*Toxoplasma gondii*, CHAG—Chagas disease, ZIKV—Zika virus, DENV—Dengue virus, RUBV—rubella virus, CMV—cytomegalovirus, HSV—herpes simplex viruses, CNS—central nervous system, IUGR—intrauterine growth restriction; ^1^ common TORCH symptoms were derived from the following sources [8,10,11].

## Data Availability

The data contained in this manuscript are available upon request.

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
