# Peer review of "TORCH Congenital Syndrome Infections in Central America’s Northern Triangle"

_microorganisms, 2023, doi:10.3390/microorganisms11020257_

Round 1

Reviewer 1 Report

Manuscript:  TORCH Congenital Syndrome Infections in Central America’s Northern Triangle

Authors: Lynn, MK, Acquino, SR, et. al.

Journal: Microorganisms

This manuscript is a review of TORCH infections in the Northern Triangle of Central America with a stated goal of identifying gaps in knowledge about these infections in maternal/infant populations in this region of Central America. Several databases, including graduate studies theses of postgraduate students from universities in this region, were searched as part of this study. As the authors note information from these sources are not peer reviewed, thus it is unclear if the findings in this manuscript can be considered a robust examination of TORCH infections in this region of Central America. However, this limitation has been acknowledged by the authors.

A major concern with this manuscript is that it allocates an inordinate amount of text describing general features of several of the individual agents associated with congenital infections without distilling critical features of maternal infections and maternal-to-fetal transmission that are unique to each agent or that can be generalized between these populations. The reader that identified this review by the title will there be somewhat misled.  In contrast, a more focused review is provided for the conventional TORCH agents (rubella, CMV etc) perhaps because more is known and much of the previous decades of work is available in the literature. As an example, the title infers this text is about congenital infections but in the Dengue section there is a discussion about Dengue vaccines but this discussion is not tailored specifically to the potential impact of these vaccines on antenatal maternal DV infections or adverse outcomes of pregnancy.  Regardless of the explanation, the first sections of the review are in my view not very informative unless the reader is willing to recover the primary references on these agents and then integrate this data into the text provided by the authors.

There are also numerous erroneous or less than informative sections and/or statements in the text such as:

1)      Line 709. No data is available so this is a review of congenital rubella syndrome without any insight into why it would be different in this population.

2)      Line 703. CMV is a betaherpesvirus  and is human herpesvirus 5 (HHV-5)

3)      Line 714. Sexual transmission is also common route.

4)      Line 729. Vertical transmission does not occur in mice

5)      Line 758. Control trials from Italy and the NIH showed no benefit of HIG

6)      Line 786. Treatment does not prevent development of latency

7)      Line 797. Viral DNA is not stored. Latency is established

8)      Line 837. There was considerable text allocated to conclude that no data is available.   

Additional examples could be listed but in general the text requires additional editing.

In summary, I believe this is an important but poorly studied area of congenital infections, primarily because this area of Central America has a high percentage of indigenous people. Unfortunately, after reading this review, I gained little new insight into how such unique populations could differ from more well studied populations in North and South America. Such a distillation of differences or similarities between maternal populations in these region of the Americas and more well described maternal populations in North/South America would be much more informative instead of re-stating many of the well known features of congenital infections.  

Author Response

This manuscript is a review of TORCH infections in the Northern Triangle of Central America with a stated goal of identifying gaps in knowledge about these infections in maternal/infant populations in this region of Central America. Several databases, including graduate studies theses of postgraduate students from universities in this region, were searched as part of this study. As the authors note information from these sources are not peer reviewed, thus it is unclear if the findings in this manuscript can be considered a robust examination of TORCH infections in this region of Central America. However, this limitation has been acknowledged by the authors.

A major concern with this manuscript is that it allocates an inordinate amount of text describing general features of several of the individual agents associated with congenital infections without distilling critical features of maternal infections and maternal-to-fetal transmission that are unique to each agent or that can be generalized between these populations. The reader that identified this review by the title will there be somewhat misled.  In contrast, a more focused review is provided for the conventional TORCH agents (rubella, CMV etc) perhaps because more is known and much of the previous decades of work is available in the literature. As an example, the title infers this text is about congenital infections but in the Dengue section there is a discussion about Dengue vaccines but this discussion is not tailored specifically to the potential impact of these vaccines on antenatal maternal DV infections or adverse outcomes of pregnancy.  Regardless of the explanation, the first sections of the review are in my view not very informative unless the reader is willing to recover the primary references on these agents and then integrate this data into the text provided by the authors.

 The article has been cut down to avoid unnecessary text devoted to infection in the general population to focus more on infection within pregnant women and congenital syndromes.

There are also numerous erroneous or less than informative sections and/or statements in the text such as:

  • Line 709. No data is available so this is a review of congenital rubella syndrome without any insight into why it would be different in this population.

The authors have chosen to remove the Rubella section from this manuscript, as it has been eliminated from the Americas in 2015 and therefore is not on the manuscripts primary theme of addressing the most salient pathogens that pose a threat to maternal and child health in this world region.

2)      Line 703. CMV is a betaherpesvirus  and is human herpesvirus 5 (HHV-5)

This sentence has been corrected.

3)      Line 714. Sexual transmission is also common route.

This has been added to the sentence regarding transmission pathways.

4)      Line 729. Vertical transmission does not occur in mice

The authors apologize for this oversight and this has been corrected.

5)      Line 758. Control trials from Italy and the NIH showed no benefit of HIG

Text was added to clarify the point that conflicting results have been obtained from HIG trials and that efficacy of this drug has not been established.

6)      Line 786. Treatment does not prevent development of latency

This sentenced was rephrased.

7)      Line 797. Viral DNA is not stored. Latency is established

This sentence was rephrased for clarity.

8)      Line 837. There was considerable text allocated to conclude that no data is available.  

This section has been cut down.

Additional examples could be listed but in general the text requires additional editing.

 In summary, I believe this is an important but poorly studied area of congenital infections, primarily because this area of Central America has a high percentage of indigenous people. Unfortunately, after reading this review, I gained little new insight into how such unique populations could differ from more well studied populations in North and South America. Such a distillation of differences or similarities between maternal populations in these region of the Americas and more well described maternal populations in North/South America would be much more informative instead of re-stating many of the well known features of congenital infections. 

Additional description of why this population is uniquely vulnerable was included to further justify the need for further studies of TORCH infections in this region.

Reviewer 2 Report

In this review article, the authors presented the most common TORCH pathogens, with an emphasis on Central America’s Northern Triangle. Please see some comments below.

Lines 53-55: Please correct toxoplasmosis, rubella, cytomegalovirus, herpes simplex, hepatitis B, and C (lowercase letters).

Table 1: I suggest using the abbreviation RUBV for rubella virus.

Please consider rephrasing "organ system conditions".

Lines 113, 121, 134: Please correct Toxoplasma gondii (in italic).

Line 210: Please correct " ... IgA antibodies"

Line 212: Please correct assess.

Lines 233-234: please correct "... the prevalence of T. gondii IgM among ..."

Lines 260, 267, 269: Please correct Trypanosoma cruzi (italic).

Line 361: Please correct "IgG serology is ..."

Lines 379, 383, 388, 400, 408: Please correct T. cruzi (italic).

Line 414: please add the abbreviation (ZIKV) and use the abbreviation in the whole paragraph consistently.

Lines 440 and 462: Please correct congenital Zika syndrome.

Line 466: Please correct ZIKV transmission.

Line 479 and 489: Please correct RT-PCR (ZIKV is an RNA virus).

Line 487: Please correct "Congenital Zika infection may ..."

Line 533: please correct RNA flavivirus.

Line 533: Please add the abbreviation Dengue virus (DENV) and use the abbreviation consistently throughout the paragraph.

Line 539: Please correct dengue virus (lowercase letter).

Line 555: Please correct mosquitoes.

Line 616: Please correct "nuclei acid amplification tests (RT-PCR) are commonly ..."

Line 623: Please correct flaviviruses.

Line 658: rubella virus should be written in a lowercase letter, please correct.

Rubella should also be written in a lowercase letter, please correct it throughout the whole paragraph.

Line 690: Please correct qRT-PCR.

Line 693: rubella is duplicated, please check.

Line 703: Please correct "Cytomegalovirus (CMV) or β-herpesvirus 5, is ..."

Line 730: please add a full stop.

Lines 745-746: The sentence "Diagnosis within the perinatal period is difficult to assess, due to the absence of symptomatic infection and the prevalence of non-primary congenital infection with CMV" is not clear. Please rephrase.

Line 781: Please correct toxoplasmosis (instead of TOXO).

Line 789: Please correct HSV-1.

References should be corrected according to the proposition of the journal.

Author Response

Reviewer 2:

In this review article, the authors presented the most common TORCH pathogens, with an emphasis on Central America’s Northern Triangle. Please see some comments below.

Lines 53-55: Please correct toxoplasmosis, rubella, cytomegalovirus, herpes simplex, hepatitis B, and C (lowercase letters).

All diseases have been changed to lower case.

Table 1: I suggest using the abbreviation RUBV for rubella virus.

The abbreviation has been changed to RUBV.

Please consider rephrasing "organ system conditions".

We have changed this symptom category for a more clear description.

Lines 113, 121, 134: Please correct Toxoplasma gondii (in italic).

Italics are present in the authors version in these lines. All mentions of T. gondii have been checked to ensure italics.

Line 210: Please correct " ... IgA antibodies"

This has been corrected.

Line 212: Please correct assess.

This has been corrected. Thank you for bringing this to our attention!

Lines 233-234: please correct "... the prevalence of T. gondii IgM among ..."

This has been corrected.

Lines 260, 267, 269: Please correct Trypanosoma cruzi (italic).

This is in italics in the authors version. The manuscript has been checked to ensure all necessary italics are present.

Line 361: Please correct "IgG serology is ..."

This has been updated per the suggestion.

Lines 379, 383, 388, 400, 408: Please correct T. cruzi (italic).

Italics have been corrected.

Line 414: please add the abbreviation (ZIKV) and use the abbreviation in the whole paragraph consistently.

The abbreviation has been updated and the section has been corrected for consistency.

Lines 440 and 462: Please correct congenital Zika syndrome.

This has been corrected.

Line 466: Please correct ZIKV transmission.

This has been corrected.

Line 479 and 489: Please correct RT-PCR (ZIKV is an RNA virus).

This has been corrected.

Line 487: Please correct "Congenital Zika infection may ..."

Corrected.

Line 533: please correct RNA flavivirus.

Flavivirus has been corrected.

Line 533: Please add the abbreviation Dengue virus (DENV) and use the abbreviation consistently throughout the paragraph.

The abbreviation has been updated here and throughout the section.

Line 539: Please correct dengue virus (lowercase letter).

This as been changed to DENV for consistency.

Line 555: Please correct mosquitoes.

This has been updated.

Line 616: Please correct "nuclei acid amplification tests (RT-PCR) are commonly ..."

This has been updated to RT-PCR.

Line 623: Please correct flaviviruses.

This has been corrected.

Line 658: rubella virus should be written in a lowercase letter, please correct.

This has been corrected throughout the section.

Rubella should also be written in a lowercase letter, please correct it throughout the whole paragraph.

Rubella has been corrected thru the whole paragraph.

Line 690: Please correct qRT-PCR.

Corrected.

Line 693: rubella is duplicated, please check.

Corrected.

Line 703: Please correct "Cytomegalovirus (CMV) or β-herpesvirus 5, is ..."

This sentence has been updated.

Line 730: please add a full stop.

This line contains a full stop, but the authors have double checked for this error.

Lines 745-746: The sentence "Diagnosis within the perinatal period is difficult to assess, due to the absence of symptomatic infection and the prevalence of non-primary congenital infection with CMV" is not clear. Please rephrase.

We have rephrased this sentence and added another sentence for clarity.

Line 781: Please correct toxoplasmosis (instead of TOXO).

This has been updated.

Line 789: Please correct HSV-1.

This typo has been corrected.

References should be corrected according to the proposition of the journal.

Reference style was updated to reflect the journal style.

Round 2

Reviewer 1 Report

Manuscript: Torch Congenital Syndrome Infections in Central America’s Northern Triangle (

Authors: Lynn, MK

Journal: Microorganisms

The revised manuscript has been improved but several statements and sections in my view require revisions. These are listed below.

1)    Line 12- is not correct and lists syndromes as infectious pathogens

2)    Line 14- Syphilis is omitted. In fact some definitions of TORCH at times were STORCH

3)    Line 61- Infective mechanisms? Not sure what this means

4)    Line 109- congenital is defined as “born with” most cases of HSV are perinatal and no symptoms are present at birth-the authors have repeatedly mixed congenital and perinatal terminology throughout the review.

5)    Line 251- 105 MB in length? Not sure of meaning of MB

6)    Line 293- DTU needs definition

7)    Line 406 – not sure of what authors are trying to describe

8)    Line 417- need to state issues of cross reactivity in serologic assays

9)    Line 499-508 – this description is ok but what does any of it have to do with congenital infection. It is filler in my view

10) Line 515-523- yes these are adverse outcomes of pregnancy but again what is the context for congenital infections?

11) Line 530-537- these are diagnostics for maternal infections- what about congenital infection, ie diagnosis of infected infant?

12) Line 540-548- context for congenital infection?

13)  Line 554-563- how strong is the case for congenital Dengue virus infections? I see two references only as in the past this was controversial

14) Line 572-what does this mean?

15) Line 583- this is not correct as there is no definitive data on the question of transmission rates in non-primary infections

16)  Line 614- there are no drugs to prevent CMV infections

17) Lines 660-671- the authors need to re-order these statements as it is really confusing whether speaking of maternal infections or perinatal infections in newborns

Overall, this manuscript still requires revision and perhaps the most obvious deficiency in this and previous versions of the manuscript is that it does not really address the most important aspect of the review, the incidence and prevalence of these infections in Central America. Repeating discussions of the natural history of these infections in other populations that can be found in more comprehensive reviews and textbooks only clouds the presentation of the data that would be of most interest to readers.  

Author Response

We thank Reviewer 1 for their second round of comments. To address their concerns, we have 1) incorporated each line-by-line edit, 2) added a concluding paragraph discussing the paper's primary contributions to the scientific literature, 3) created a supplementary table to highlight the regional specific TORCH studies, and 4) revised the overall paper to focus less on general knowledge (e.g. pathogen life cycles) and more on the unique epidemiologic studies. 

We hope these continued edits satisfy the reviewer. Thank you.  

Round 3

Reviewer 1 Report

revision has addressed concerns raised in the initial reviews